# New Haloterpenes from the Marine Red Alga *Laurencia papillosa*: Structure Elucidation and Biological Activity

**DOI:** 10.3390/md19010035

**Published:** 2021-01-14

**Authors:** Mohamed Shaaban, Ghada S. E. Abou-El-Wafa, Christopher Golz, Hartmut Laatsch

**Affiliations:** 1National Research Centre, Chemistry of Natural Compounds Department, Division of Pharmaceutical Industries, El-Behoos St. 33, Dokki, Cairo 12622, Egypt; 2Institute of Organic and Biomolecular Chemistry, University of Göttingen, Tammannstrasse 2, D-37077 Göttingen, Germany; cgolz@gwdg.de; 3Department of Botany, Faculty of Science, Mansoura University, Algomhuria st. 60, El-Mansoura 35516, Egypt; ghadas@mans.edu.eg

**Keywords:** marine red algae, *Laurencia papillosa*, haloterpenes, biological activity

## Abstract

Analysis of the air-dried marine red alga *Laurencia papillosa*, collected near Ras-Bakr at the Suez gulf (Red Sea) in Egypt delivered five new halogenated terpene derivatives: aplysiolic acid (**1**), 7-acetyl-aplysiol (**2**), aplysiol-7-one (**3**), 11,14-dihydroaplysia-5,11,14,15-tetrol (**5a**), and a new maneonene derivative **6**, named 5-*epi*-maneolactone. The chemical structures of these metabolites were characterized employing spectroscopic methods, and the relative and absolute configurations were determined by comparison of experimental and ab initio-calculated NMR, NOE, ECD, and ORD data, and by X-ray diffraction of **2** and **6**. The antimicrobial activities of the crude extract and compounds **1**–**3**, **5a** and **6** were studied.

## 1. Introduction

Among the red algae, the genus *Laurencia* is known to produce the largest number and diversity of secondary metabolites, making it the world’s chemically most complex seaweed genus [1]. Furthermore, several *Laurencia* metabolites have exhibited noteworthy antibacterial [2,3,4,5] insecticidal [3,6], antifungal [7], and antiviral activity [8], in addition to their anti-inflammatory, antiproliferative, antifouling, antifeedant, cytotoxic, ichthyotoxic, and insecticidal properties [9]. Certain *Laurencia* species have shown to be an unprecedented source of halogenated secondary metabolites, predominantly sesquiterpenes [10], diterpenes [11], and C_15_ non-terpenoids, including halogenated cyclic enyne ethers and related allenes [12,13,14]. An example is the halogenated diterpene aplysiadiol (**4**), which was obtained from the marine mollusk *Aplysia kurodai* [15] and a Malaysian *Laurencia* species, and exhibited potent antibacterial activities [16].

In the course of our research to isolate and investigate bioactive metabolites from *Laurencia* algae collected from the shallow waters of the Red Sea on Egyptian coasts, a series of four halogenated bioactive decalin derivatives was isolated, namely aplysiolic acid (**1**), 7-acetyl-aplysiol (**2**), aplysiol-7-one (**3**), 11,14-dihydroaplysia-5,11,14,15-tetrol (**5a**), and a further chloro compound **6** (Figure 1). We named the latter 5-*epi*-maneolactone (**6**), because of the similarity with the maneonenes, bioactive C_15_ acetogenins from *L. obtusa* [17,18]. Additionally, thyrsiferol [19], 10-hydroxykahukuene B [20], cholesterol, hexadecanoic acid, thymine, and uracil were gained from the dichloromethane-methanol extract of *L. papillosa*. The chemical structures of these compounds were determined by extensive 1D and 2D NMR and ESI HR mass measurements. Their relative and absolute configurations were derived by 1D and 2D NMR measurements, from ab initio calculations of ECD, ORD, ^13^C NMR data, by geometry-derived NOE predictions, and by X-ray diffraction in case of **2** and **6**. The antimicrobial activity of the crude extract and of the new compounds was studied using a set of diverse microorganisms.

## 2. Results and Discussion

### 2.1. Working up and Structure Identification

The dichloromethane-methanol extract of air-dried *L. papillosa* algae was fractionated by a series of chromatographic purification steps implying silica gel and size exclusion columns, to afford the haloterpenes **1**–**3**, **5a**–**c**, and **6** (Figure 1), and additionally thyrsiferol [19], 10-hydroxykahukuene B [20], cholesterol, hexadecanoic acid, thymine, and uracil. The physico-chemical properties of the new compounds are listed in Table 1.

#### 2.1.1. Aplysiolic Acid

Compound **1** was isolated as an optically active colorless solid, exhibiting neither UV-absorbance nor fluorescence on silica gel. On TLC, it was detected as a pink spot after spraying with anisaldehyde/sulfuric acid, tentatively pointing to a terpenoidal moiety. The (−)-ESI mass spectrum showed [M − H]^–^ ion signals of nearly equal intensities at *m/z* 301 and 303 Dalton, and a triple ion-peak at *m/z* 605 [2M − H]^−^, indicating the presence of one bromine atom. The (−)-ESI HR mass spectrum suggested the molecular formula C_13_H_19_BrO_3_, indicating four double bond equivalents (DBE) (Table 1).

The ^13^C NMR spectrum (Table 2) showed a total of 13 resonances, amongst them ten in the aliphatic, two in the olefinic, and one in the carbonyl region. The ^1^H NMR pattern exhibited two singlets at *δ* 4.90 and 4.81, which were assigned by H,H COSY and HMQC spectra as exo-methylene protons (*δ*_C_ 110.2). The respective quaternary carbon signal appeared at *δ* 148.5 and was assigned by HMBC correlations (Figure 2). The carbonyl group, together with one methyl, six methylene, and two methine groups and three fully substituted carbons, sums up to C_13_H_17_BrO. Respectively, **1** should contain two rings, and the remaining two hydrogen and oxygen atoms are forming hydroxyl groups.

Based on the H,H COSY correlations, two fragments were identified in **1**, a 1,1,3-trisubstituted propane (-CH_2_-CH_2_-CH-), and a 1,2,4-trisubstituted butane (-CH_2_-CH-CH_2_-CH_2_-) unit, respectively. The terminal methylene protons in the first fragment (*δ* 2.70, 2.15) showed ^2^*J* and ^3^*J* HMBC correlations with the olefinic carbons C-4 and C-13, confirming their direct neighborhood. One of these methylene protons (*δ* 2.70) showed a ^4^*J* COSY signal with the exo-methylene protons. The methine proton H-1 (*δ* 4.72) exhibited ^2^*J* and ^3^*J* correlations towards C-10 (*δ* 43.0) and the methyl singlet of C-12 (*δ* 14.8), respectively, suggesting a connection of C-10 with C-1 and Me-12. If C-5 is hydroxylated (*δ* 76.2), the downfield shift of C-1 (*δ* 62.9) fits best on its substitution with bromine [21]. Among others, the methylene group CH_2_-3 showed a ^3^*J* HMBC correlation towards C-5, which itself gave cross signals with the methylene protons H_2_-13 and Me-12, so that an exomethylene-cyclohexane (**ring A**) was formed.

Further HMBC and NOESY correlations (Figure 2 and Figure 3) indicated that the butanediyl fragment formed a second cyclohexane (**ring B**) via C-10 (*δ* 43.0) and C-5 (*δ* 76.2). CH_2_-9 correlated with C-10 and Me-12 and showed all other expected HMBC and NOESY correlations. In a similar way, the ring closure between CH_2_-6 and C-5 was confirmed. According to ^2^*J* and ^3^*J* HMBC correlations from H_ax,β_-6 (*δ* 2.04), H_α_-7 (*δ* 2.94), and H_ax,β_-8 (*δ* 1.66, weak) with C-11 (*δ* 181.0), the missing atoms are forming a carboxyl group at C-7.

Due to the low flexibility of **1**, its NOESY signals were surprisingly clear and easy to interpret. Me-12 showed correlations with H_ax,β_-2, H_ax,β_ -6, H_ax,β_-8, and H_eq,β_-9 but not with H-1, H_α_-2, H_α_-6, H_α_-7, and H_α_-8, so that all correlating protons should be in a *syn*-facial position with the methyl group. The NOE between H-7 and H_eq,α_-6, H_eq,α_-8, H_ax,α_-9, and of H-1 with H_eq,α_-2, H_ax,α_-3, H_ax,α_-9, but not with Me-12, indicated their anti-position with respect to Me-12, resulting in the rel-(1*S*,7*S*,10*S*)-configuration. DFT-calculations of the **1**-geometry showed that the strong NOE effect between (*Z*)-H-13 (*δ* 4.81) and H_2_-6 (*δ* 2.04, 1.83), and between H_β_-6 (2.04) and Me-12 (0.96) is not explainable by a *cis*-decalin, but requires definitely a *trans*-decalin with a rel-(5*S*,10*S*) configuration (see Appendix A). As the sign of the ORD data calculated for all-(*S*)-**1** agreed with the experimental data, the decalin was elucidated as 5-bromo-8a-hydroxy-4a-methyl-8-methylene-decahydronaphthalene-2-carboxylic acid (**1**), having the absolute all-(*S*)-configuration. We named it aplysiolic acid, with respect to the similarity with aplysiadiol (**4**) [15]. For aplysiadiol (**4**) and anhydroaplysiadiol [21], the same relative configuration as for **1** was published, and we confirmed their absolute configuration by agreement of calculated and published negative sign of their optical rotation as all-(*S*) as well (Appendix A).

#### 2.1.2. 7-Acetyl-aplysiol

An additional optically active colorless solid had similar chromatographic properties as **1**, but with noticeably less polarity. The ESI HR mass spectrum indicated the molecular formula C_14_H_21_BrO_2_ with four double bond equivalents (DBE) as well. However, an OH group in **1** was formally exchanged against a methyl group. Respectively, a methyl singlet (δ_H_ 2.13) and its corresponding carbon (28.3) were visible. The carboxyl signal in **1** (181.0) was exchanged in **2** against a keto carbonyl at δ_C_ 211.8, but the remaining ^1^H and ^13^C NMR signals were very similar in both compounds. On the basis of long-range 2D NMR couplings (Figure 2) and NOESY experiments (Figure 4), the structure was assigned as **2**, which we named 7-acetyl-aplysiol. From this ketone, the acid **1** may be formed in a haloform reaction (Einhorn reaction). With respect to the negative optical rotation, the decalin system of **2** should have the same absolute all-(*S*)-configuration as **1**, which also agrees with biosynthetic considerations. However, as some doubt remained after comparison of the calculated and experimental ECD spectra, a crystal of **2** was analyzed additionally by X-ray diffraction (Appendix A), which reassured the all-(*S*)-configuration.

#### 2.1.3. Aplysiol-7-one

A third closely related decalin gave the molecular formula C_12_H_17_BrO_2_ by ESI HR, containing 4 DBE as well. The ^1^H and ^13^C NMR spectra of **3** resembled those of **1** and **2**; however, the signals of CH-7 (in **1** and **2**) and CH_3_-14 (in **2**) were absent, and the ABX signal of CH_2_-6 in **1** and **2** had changed into an AB signal. As indicated by the HMBC correlations and the downfield shifts of CH_2_-6 and CH_2_-8, the carbonyl signal at δ_C_ 209.0 was due to a ß-decalone. Furthermore, long-range 2D NMR couplings (Figure 2) confirmed structure **3**, which we named as aplysiol-7-one. In addition, the all-(*S*)-configuration was plausible for NOESY experiments (Figure 4) and biosynthetic reasons; it was, however, not further analyzed due to an inseparable contamination by 10-hydroxykahukuene B [20].

#### 2.1.4. 11,14-Dihydroaplysia-5,11,14,15-tetrols 

A further brominated terpenoide with moderate polarity was isolated as an optically active resin with color reactions similar as of **1** (Table 1). The NMR spectrum (Table 3) of this compound showed 20 ^13^C signals. At high magnification, however, each of them appeared as a group of up to four signals in distances of <0.2 ppm (Appendix A), so that a mixture of four stereoisomers or otherwise closely related compounds was expected; this was confirmed by the ^1^H NMR signal of the Δ^12^ double bond (Appendix A) and also by analytical HPLC analyses, where three components in the ratio of ~1:1:1 were separated. By HPLC/HRMS, two stereoisomers C_20_H_33_BrO_4_ (**5a**,**b**) and a slightly more polar component C_20_H_33_BrO_5_ (**5c**) were detected (Appendix A). The latter compound seems to be a 11,14-dihydroaplysiapentol; we need, however, to isolate further material to fully elucidate the structure (see Appendix A).

Both molecular formulas indicated four DBE (Table 1), and the COSY, HSQC, HMBC (Figure 2), and NOESY (Appendix A) data confirmed the same brominated rel-(1*S*,5*S*,7*S*,10*S*)-exomethylene-decalin skeleton as in **1**–**3**. However, instead of the carboxy or the acetyl group, respectively, a 6-methyl-hept-3-ene-2,5,6-triol-2-yl side chain was found: the singlet of Me-18 (δ_H_ 1.27) displayed HMBC correlations with C-7, C-11 (δ_C_ 75.4), and the olefinic carbon C-12 (δ_C_ 139.8). Further COSY and HMBC correlations completed the chain from CH-7 to Me-16/Me-17. Accordingly, structures **5a/b** were elucidated as further new aplysiadiol derivatives [15] and named, respectively, as 11,14-dihydroaplysia-5,11,14,15-tetrols. Also here, the same absolute configuration as in **1**–**3** was assumed for the decalin system for biosynthetic reasons, so that only the stereochemistry of the side chain remained open. On this basis, the absolute configuration of the main component **5a** was determined as (1*S*,5*S*,7*S*,10*S*,11*R*,14*R*) by correlation of NOE signals with *ab-initio* calculated atom distances (Appendix A); the other diastereomers could not be assigned, due to overlapping signals.

Aplysiadiol derivatives are biosynthetically regarded as extended sesquiterpenes; they represent examples of the rare prenylated eudesmane type, which commonly occur in terrestrial plants [22]. A few compounds of this type have been isolated from marine mollusks [15], brown algae [23], and soft corals [24,25,26]. 

#### 2.1.5. 5-epi-Maneolactone

Compound **6** displayed neither UV absorbance nor fluorescence on TLC, but turned brownish gray with anisaldehyde/sulfuric acid on heating. The molecular weight was determined by ESI MS: two *quasi*-molecular ion peaks in the ratio of 1:0.33 at *m/z* 238 and 240 indicated the presence of one chlorine atom. The corresponding molecular formula C_12_H_11_ClO_3_ was determined by ESI HRMS and indicated seven DBE (Table 1). The IR spectrum of **6** displayed a characteristic vibration signal (*ν* = 2361 cm^−1^) of an acetylenic bond. Two absorption bands at *ν* =1781 and 1592 cm^−1^ indicated the presence of a lactone carbonyl and an olefinic double bond, respectively, so that three rings were expected.

The ^1^H NMR methine signal at δ_H_ 3.36 did not show an HSQC correlation; however, a large HMBC coupling with the CH carbon at δ_C_ 86.5 and a smaller one with a C_q_ at δ_C_ 78.2 was noted. Together with strong long-range correlations with a *cis*-double bond (δ_C_ 113.8, 139.2, *J* = 10.5 Hz), this indicated a terminal acetylene in conjugation with a double bond. According to further HMBC and COSY correlations, the (*Z*)-enyne unit was connected with the methines C-5 and C-6, and the latter additionally with CH-7 and CH-11 (Figure 5). A detailed analysis resulted in a ((*Z*)-pent-2-en-4-ynyl)-cyclohexane, where the shift assignments of positions CH-7, -9, and -10 remained open due to their overlapping ^1^H NMR signals (Table 4).

According to AntiBase [27], only two groups of metabolites with these structural features have been described before, the lembynes [2,28], and the maneonenes [17,18]. All were isolated from *Laurencia* spp. If chlorine was present, it was found in position C-5; also the shifts of **6** were explained best by chlorine at C-5. HMBC correlations of the upfield methine protons CH-6 and CH-11 with the signal at δ_C_ 173.5 connected the ester carbonyl C-12 with C-11, and the correlation with the oxymethine CH-9 (or CH-7) via oxygen closed a lactone ring with C-12. The remaining oxygen atom bridges the oxymethine groups CH-10 with CH-7 or CH-9, so that all DBE were used. Further correlation analyses by means of COCON [29] delivered four structure options, with structure **6** being the only plausible one (Figure 5 and Appendix A). A suitable crystal allowed us finally to confirm this structure including the absolute configuration unambiguously by X-ray diffraction (Figure 6). DFT calculations afforded the same absolute configuration, so that the validity of the computational methods applied here were additionally confirmed (see Appendix A). We named compound **6** as 5-*epi*-maneolactone, as it looks like an oxidative cleavage product of the *cis*-maneonenes. It should be mentioned, however, that C-5 is (*R*)-configured in these compounds, oppositely to the (5*S*)-configured **6**.

### 2.2. Biological Activities

Antimicrobial activity testing of the crude extract of the red alga *L. papillosa* was carried out against ten microorganisms using the agar diffusion technique with paper platelets: *Bacillus subtilis* ATCC6051, *Staphylococcus aureus*, *Streptomyces viridochromogenes* Tü 57, *Streptococcus pyogenes*, *Escherichia coli*, *Shigella* sp., *Proteus* sp., *Candida albicans*, *Mucor miehei* Tü 284, and *Chlorella vulgaris*. The crude extract showed at 400 μg/disk a strong antibacterial activity against the Gram-positive *Streptomyces viridochromogenes* Tü 57 (30 mm). Among the tested compounds **1**–**3**, **5**, and **6**, aplysiolic acid (**1**) and aplysiol-7-one (**3**) were only moderately active against *S. aureus* (10.5 mm) at a concentration of 40 μg/disk.

## 3. Materials and Methods

### 3.1. General Procedures

IR spectra: FT/IR-4100 Infrared Spectrometer (Jasco, Inc., Easton, MD, USA). NMR spectra were measured with Varian Unity 300 and Varian Inova 600 spectrometers (Varian, Palo Alto, CA, USA). Optical rotations: Polarimeter (Perkin-Elmer, model 343, Perkin-Elmer Life and Analytical Sciences, Shelton, CT, USA). Electron spray ionization mass spectrometry (ESI MS): Finnigan LCQ ion trap mass spectrometer coupled with a Flux Instruments (Basel, Switzerland) quaternary pump Rheos 4000 and an HP 1100 HPLC (Nucleosil column EC 125/2, 100-5, C 18) with autosampler (Jasco 851-AS, Jasco, Inc., Easton, MD, USA) and a diode array detector (Finnigan Surveyor LC System, San Jose, CA, USA). High-resolution mass spectra (HRMS) were recorded by ESI MS on a MicrOTOF mass spectrometer (Bruker Daltonics, Billerica, MA, USA) or a LTQ Orbitrap XL (Thermo Fisher Scientific wissenschaftliche Geräte GmbH, 20354-Hamburg, Germany), equipped with an HPLC instrument (Thermo Scientific Accela HPLC System Markham, Ontario, Canada); for further details, see mass spectra in Appendix A, e.g., Appendix A. Rf values are listed in Table 1 and were determined on Polygram SIL G/UV254 (Macherey & Nagel, Düren, Germany). Size exclusion chromatography was performed on Sephadex LH-20 (Lipophilic Sephadex; Amersham Biosciences, Ltd., purchased from Sigma-Aldrich Chemie, Steinheim, Germany).

### 3.2. Collection and Taxonomy of the Marine Alga

The red alga *L. papillosa* (Forsk., Grev) was collected in summer 2009 at Ras Abu-Bakr, 65 km north of Ras Gharib on the Suez-Gulf, Red Sea, Egypt. The identification was carried out by G. S. Abou-El Wafa according to Nasr’s method [30,31]. A reference specimen of the alga is kept at the Department of Botany, Faculty of Science, Mansoura University, Egypt. Samples of *L. papillosa* were separated from epiphytes and the algal material rinsed in tap water and distilled water. The sample was then spread on string nets, allowed to dry in air, ground, and stored in closed bottles at room temperature.

### 3.3. Extraction and Isolation of the Bioactive Constituents

The air-dried algal sample (~850 g) was extracted at room temperature with dichloromethane/methanol (1:1, 48 h for each batch). The extract was concentrated under reduced pressure to gain 8.83 g of residue, which was applied to column chromatography on silica gel (SC) using a gradient elution starting first with cyclohexane, then cyclohexane-CH_2_Cl_2_ and finally with CH_2_Cl_2_-MeOH. By TLC monitoring (UV light 254 and 366 nm, Merck, 64579 Gernsheim, Germany) and by using anisaldehyde/sulfuric acid spray reagent, five fractions were obtained: I (1.73 g), II (0.32 g), III (0.73 g), IV (1.86 g), and V (1.92 g). Further SC (silica gel, cyclohexane-CH_2_Cl_2_) of fraction I afforded cholesterol (46 mg) and hexadecanoic acid (66.7 mg) as colorless solids. Purification of fraction II (SC on silica gel eluted with a cyclohexane/DCM/MeOH gradient and subsequently Sephadex LH20 (CH_2_Cl_2_/40% MeOH) afforded compound **5** as a pale-yellow oil (165 mg) and **6** as a colorless solid (6 mg), respectively. SC of fraction IV on silica gel with a DCM/MeOH gradient and subsequently Sephadex LH20 (CH_2_Cl_2_/40% MeOH)) resulted in 7-acetyl-aplysiol (**2**, 19.4 mg), aplysiol-7-one (**3**, 4.8 mg) and 10-hydroxykahukuene B (5 mg) as colorless solids. SC of fraction V on silica gel (CH_2_Cl_2_/MeOH), followed by PTLC (DCM/10% MeOH) and Sephadex LH20 (MeOH) delivered aplysiolic acid (**1**, 1.4 mg), thyrsiferol (1.5 mg), uracil (1 mg), and thymine (1 mg) as colorless solids. Detailed 2D NMR correlations (H,H COSY and HMBC) of compounds **1**–**3**, **5a** and **6** have been given in the Appendix A.

### 3.4. Antimicrobial Assay

Antimicrobial assays using the agar diffusion test [32] were performed as described previously [33]. *M. miehei* Tü 284 and *S. viridochromogenes* Tü 57 were obtained from the collection of H. Zähner (University of Tübingen, Germany), and *Chlorella vulgaris* was provided by the Algal Collection Göttingen. *B. subtilis* ATCC 6051 was obtained from the American Type Culture Collection, while *S. aureus*, *E. coli* and *C. albicans* are clinical isolates from Göttingen hospitals. Strains are kept in the strain collection of H. Laatsch, Institute of Organic and Biomolecular Chemistry, Georg-August University, Göttingen, Germany.

### 3.5. Ab Initio Calculations 

DFT calculations were performed as described previously [34].

### 3.6. Crystal Structure Determination of 7-acetylaplysiol (2) and 5-epi-maneolactone (6)

The structures of **2** and **6** were determined by single-crystal X-ray diffraction on two dual source equipped Bruker D8 Venture four-circle-diffractometer (Bruker AXS GmbH, Karlsruhe, Germany). 7-Acetylaplysiol (**2**), C_14_H_21_BrO_2_, was crystallized from methanol as yellow plates in the orthorhombic crystal system in the non-centrosymmetric cpase group *P*2_1_2_1_2_1_; 5-*epi*-maneolactone (**6**), C_12_H_11_ClO_3_, was crystallized from CHCl_3_/20% MeOH as colorless blocks in the monoclinic crystal system in the non-centrosymmetric space group *P*2_1_. Absolute configuration could be determined reliably for both compounds with Flack’s parameter of 0.001(5) and −0.014(7) for 7-acetylaplysiol (**2**) and 5-*epi*-maneolactone (**6**), respectively. Full crystallographic information can be retrieved from the CIF file and the Appendix A. CCDC 2041564 (**2**) and 2008525 (**6**) contain the supplementary crystallographic data for this paper. These data can be obtained free of charge from The Cambridge Crystallographic Data Centre via www.ccdc.cam.ac.uk/data request/cif. For further data, refer to the Appendix A.

## 4. Conclusions

Six haloterpenes, namely aplysiolic acid (**1**), 7-acetyl-aplysiol (**2**), aplysiol-7-one (**3**), 11,14-dihydroaplysia-5,11,14,15-tetrol (**5a**), the epimer **5b** and 5-*epi*-maneolactone (**6**), along with thyrsiferol, 10-hydroxykahukuene B, cholesterol, hexadecanoic acid, thymine, and uracil were isolated from the marine red alga *L. papillosa*. The chemical structures of the new metabolites were characterized by employing spectroscopic methods (1D, 2D NMR, and ESI HR mass measurements). The relative and absolute configurations of the new compounds were determined by ab initio calculations of ECD, ORD, and NMR data, and for **2** and **6**, additionally by X-ray diffraction. In a set of microorganisms, the crude extract was strongly active against *Streptomyces viridochromogenes* Tü 57 (30 mm), whereas the activity of the pure metabolites was low.

## Figures and Tables

**Figure 1 marinedrugs-19-00035-f001:**
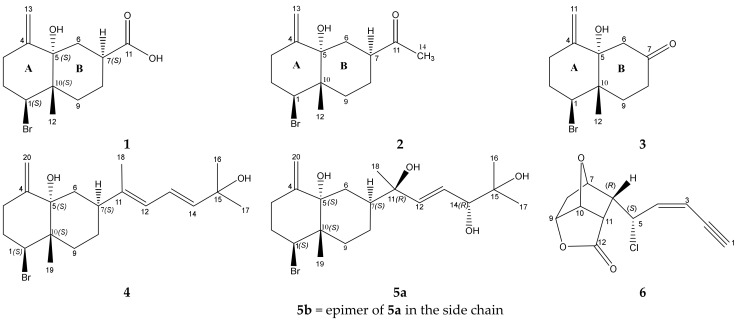
Structures of new haloterpenes **1**–**3**, **5a**, **6** from *L. papillosa**.***

**Figure 2 marinedrugs-19-00035-f002:**
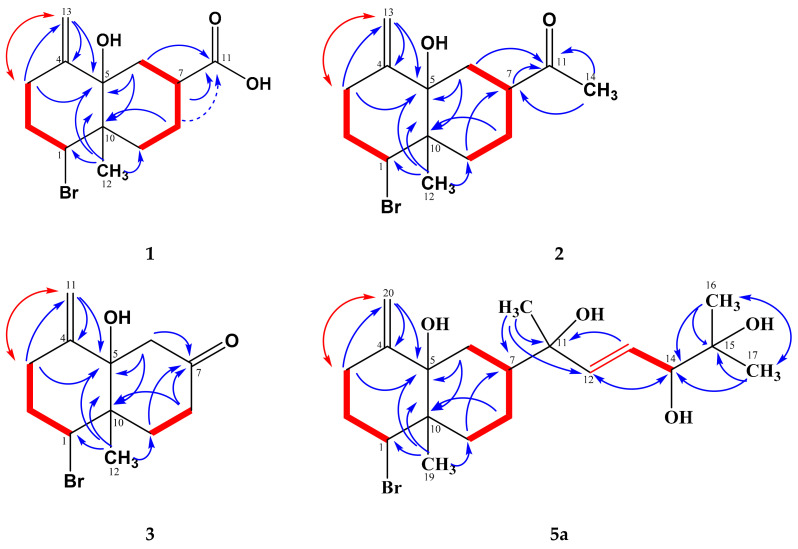
H,H COSY (^3^*J*


, ^4^*J*

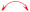
) and HMBC (

, 
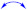
) correlations of **1**–**3, 5a**.

**Figure 3 marinedrugs-19-00035-f003:**
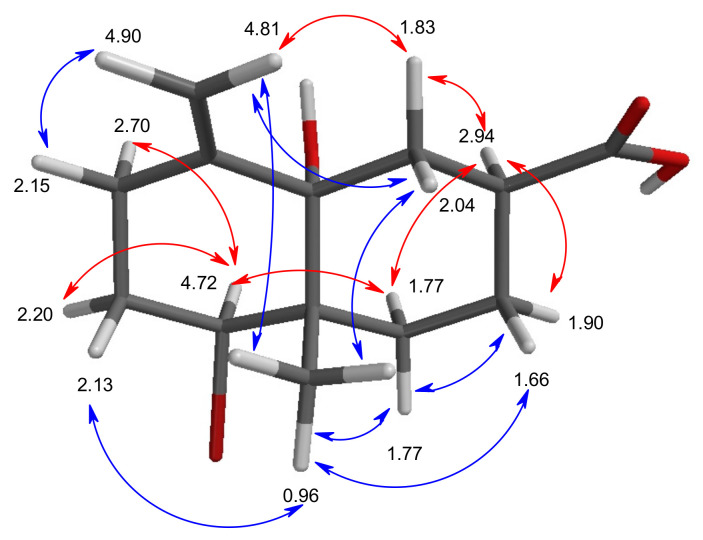
NOESY correlations (
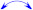
 = front side, β-orientation, 
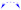
 = weak); 
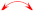
 = backside, α-orientation;) of all-(*S*)-aplysiolic acid (**1**); proton shifts = values at the atoms. For atom distances, see Appendix A.

**Figure 4 marinedrugs-19-00035-f004:**
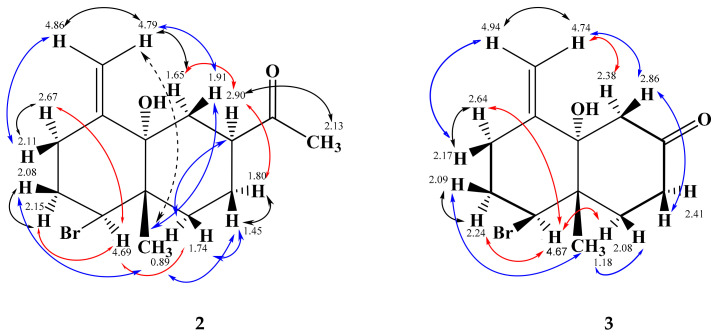
Main NOESY correlations (

 = front side, *β*-orientation 

 = weak; 
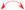
 = backside, α-orientation; 
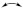
 geminal correlations) of all-(*S*)-7-acetyl-aplysiol (**2**) and all-(*S*)-aplysiol-7-one (**3**).

**Figure 5 marinedrugs-19-00035-f005:**
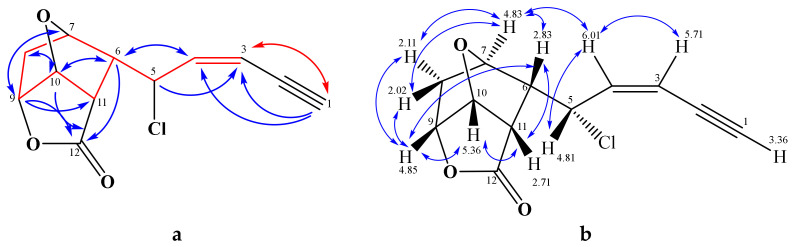
(**a**) H,H COSY (



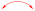
) and selected HMBC (

) correlations of 5-*epi*-maneolactone (**6**, left), (**b**) NOESY connectivities (
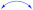
) of 5-*epi*-maneolactone (**6**, right).

**Figure 6 marinedrugs-19-00035-f006:**
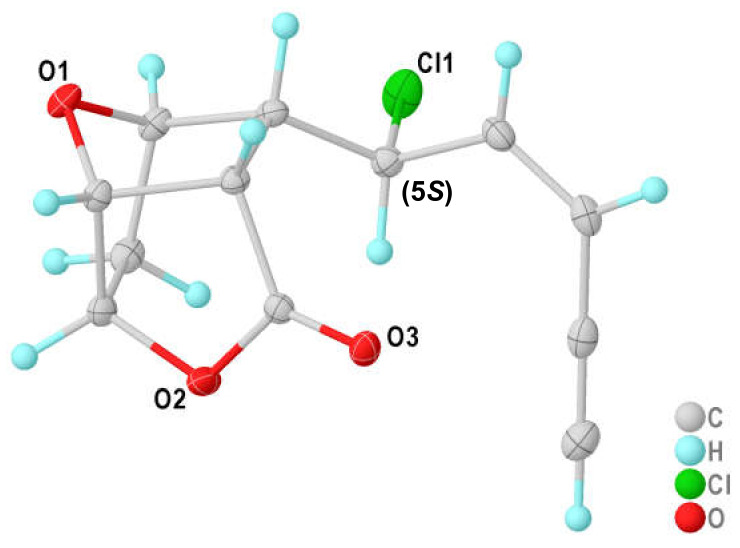
Crystal structure and absolute configuration of *5-epi-*maneolactone (**6**) by X-ray diffraction.

**Table 1 marinedrugs-19-00035-t001:** Physico-chemical properties of compounds **1**–**3a**–**c**, **5a** and **6**.

	Aplysiolic Acid (1)	7-Acetyl-aplysiol (2)	Aplysiol-7-one (3)	Dihydroaplysiatetrol (5a)	5-*epi*-Maneolactone (6)
Appearance	colorless solid	colorless solid	colorless solid	colorless oil	colorless solid
*R* _f_	0.26 ^a^	0.28 ^b^	0.39 ^b^	0.30 ^a^	0.50 ^b^
Anisaldehyde/ sulfuric acid	pink, turning later to violet	pink, turning later to violet	pink, turning later to violet	pink, turning later to violet	brownish gray
Molecular formula	C_13_H_19_BrO_3_	C_14_H_21_BrO_2_	C_12_H_17_BrO_2_	C_20_H_33_BrO_4_	C_12_H_11_ClO_3_
(+)-ESIMS: *m/z* (%)		323/325 [M + Na]^+^ (100:95.4), 623/625/627 [2M + Na]^+^, (10:31:10)	296/298 [M + Na]^+^ (88:100), 569 [2M + Na]^+^	439/441 [M + Na]^+^ (100:97), 855/857/859 [2M + Na]^+^, (20:57:19)	261/263 [M + Na]^+^ (100:31), 499 [2M + Na]^+^ (4)
(−)-ESIMS: *m/z* (%)	301/303 [M − H]^–^ (100:90.5), 603/605/607 [2M − H]^–^(37:57:21)	335/337/339 [M + Cl]^−^ (69:100:25)		461/463 [M + 2Na − 3H]^−^ (100:97), 831/833/855 [2M − H]^–^, (25:52:27)	
(+)-ESIHRMS: (*m/z*)		323.0615 [M + Na]^+^ (calc. 323.0617 for C_14_H_21_BrNaO_2_), 625.1326 [2M + Na]^+^ (calc. 625.1327 for C_28_H_42_Br_2_NaO_4_)		439.1454 [M + Na]^+^ (calc. 439.1454 for C_20_H_33_BrO_4_Na)	261.0299 [M + Na]^+^ (calc. 261.0289 for C_12_H_11_ClO_3_Na)
(−)-ESIHRMS (*m/z*)	301.0441 [M − H]- (calc. 301.0444 for C_13_H_18_BrO_3_)	335.0418 [M + H]^–^ (calc. 335.0418 for C_14_H_21_BrClO_2_)			
IR (KBr) *ν* cm^−1^					3259, 2361, 2182, 1782, 1592, 1358, 1160, 1018, 895, 841, 794, 665
[α]^20^_D_ (MeOH)	−34.8 (c, 0.13)	−55.4 (c, 0.24)		− 31.9 (c, 0.12)	−155.8 (c, 0.26)

^a^ CH_2_Cl_2_/10% MeOH; ^b^ CH_2_Cl_2_/3% MeOH.

**Table 2 marinedrugs-19-00035-t002:** ^13^C (125 MHz) and ^1^H (600 MHz) NMR data of aplysiolic acid (**1**) and related analogues **2**, **3** in CDCl_3_.

Nr.	1		2		3	
	δ_C_	δ_H_ (*J* in Hz)	δ_C_	δ_H_ (*J* in Hz)	δ_C_	δ_H_ (*J* in Hz)
1	62.9	4.72 (dd, 12.4, 4.7)	63.2	4.69 (dd, 12.3, 4.7)	61.5	4.67 (dd, 12.7, 4.5)
2	33.9	2.20 (m), 2.13 (m)	33.9	2.15 (m), 2.08 (m)	34.0	2.24 (m), 2.09 (m)
3	32.5	2.70 (dddd, 13.5, 9.6, 5.7, 2.1)2.15 (m)	32.5	2.11 (m), 2.67 (tdt, 13.0, 5.6, 2.0)	32.1	2.64 (ddd, 13.6, 5.0, 2.8), 2.17 (ddd, 13.6, 5.0, 2.0)
4	148.5		148.6		147.7	
5	76.2		76.3		79.6	
6	34.1	2.04 (dd, 13.8, 12.8), 1.83 (m)	33.1	1.65 (ddd, 14.0, 3.9, 1.3), 1.91 (m)	48.5	2.86 (d, 14.6), 2.38 (m)
7	38.2	2.94 (tt, 12.9, 4.1)	46.3	2.90 (tt, 12.7, 4.0)	209.9	
8	23.5	1.90 (m), 1.66 (m)	23.4	1.80 (m), 1.45 (m)	37.5	2.41 (m)
9	32.0	1.77 (m)	32.2	1.74 (m)	33.3	2.08 (m)
10	43.0		43.0		43.4	
11	181.0		211.8		110.9	4.94 (d, 2.0), 4.74 (d, 1.5)
12	14.8	0.96 (s)	14.7	0.89 (s)	14.8	1.18 (s)
13	110.2	4.90 (dd, 2.0, 1.0), 4.81 (d, 1.5)	110.0	4.86 (m), 4.79 (t, 1.3)		
14	-		28.3	2.13 (s)		

“-” means that there is no carbon and therefore also no shift, as seen in the structure. An alternative is just to omit the hyphen.

**Table 3 marinedrugs-19-00035-t003:** ^13^C (125 MHz) and ^1^H NMR (300 MHz) data of dihydroaplysiatetrols (**5a** and **5b**) in CD_3_OD. For the ^13^C shifts, we used two digits behind the decimal point to differentiate between **5a** and its epimer **5b**); in Appendix A, the average of shifts for **5a** and **5b** was used with only one digit.

Nr.	5a		Epimer 5b	
	δ_C_	δ_H_ (*J* in Hz)	δ_C_	δ_H_ (*J* in Hz)
1	65.23	4.72 (m)	65.23	4.72 (m)
2	35.61	2.14 (m), 2.10 (m)	35.61	2.13 (m), 2.08 (m)
3	33.69	2.77 (m), 2.12 (m)	33.69	2.77 (m), 2.09 (m)
4	151.64		151.61	
5	77.28		77.27	
6	32.75	1.66 (m), 1.75 (m)	32.75	1.66 (m), 1.73
7	43.48	1.95 (m)	43.65	1.95 (m)
8	22.22	1.33, 1.67 (m)	22.93	1.33, 1.60 (m)
9	34.03	1.68 (m)	33.96	1.67 (m)
10	44.17		44.17	
11	75.39		75.38	
12	139.83	5.79 (d, 16.3)	140.46	5.77 (d, 15.6)
13	128.08	5.74 (dd, 15–16, 6–7)	127.95	5.71 (dd, 15–16, 6–7)
14	80.33	3.85 (m)	80.39	3.86 (m)
15	73.69		73.66	
16	25.97	1.15 (s)	26.07	1.15 (s)
17	25.11	1.15 (s)	25.16	1.16 (s)
18	26.30	1.27 (s)	26.22	1.27 (s)
19	15.21	0.89 (s)	15.21	0.891 (s)
20	109.15	4.84 (s), 4.74 (s)	109.17	4.83 (s), 4.75 (s)

**Table 4 marinedrugs-19-00035-t004:** ^13^C (125 MHz) and ^1^H NMR (300 MHz) data of 5-*epi*-maneolactone (**6**) in CDCl_3_.

Nr.	δ_C_	δ_H_ (*J* in Hz)
1	86.5	3.36 (dd, 2.4, 1.0)
2	78.2	
3	113.8	5.71 (ddd, 10.5, 2.5, 0.5)
4	139.2	6.01 (ddd, 10.5, 9.5, 1.1)
5	55.3	4.81 (m)
6	52.9	2.83 (dddd, 12.1, 10.4, 4.1, 1.7)
7	80.0	4.83 (m)
8	33.9	2.11 (d,18.0), 2.02 (m)
9	78.9	4.85 (m)
10	84.1	5.36 (td, 5.0, 0.9)
11	42.9	2.71 (ddd, 10.2, 5.1, 0.8)
12	173.5	

## Data Availability

Further data are available on request from the corresponding authors.

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
