# Peer review of "New Haloterpenes from the Marine Red Alga *Laurencia papillosa*: Structure Elucidation and Biological Activity"

_marinedrugs, 2021, doi:10.3390/md19010035_

Round 1

Author Response

Comments and suggestions for improvement:

  1. General comments This manuscript was well written and well documented. It was very fortunate that the crystal structures were solved for two of the compounds. This made it easier to verify the assignments of the chiral centers, configurations and other aspects of the structures. The extensive spectra and tables in the supplementary materials were very helpful to generate confidence that the structures were correctly identified. Most of the questions I had while reading this paper were answered in subsequent paragraphs.

Response 1. Thank you very much for your support and for the helpful comments and suggestions, which are clearly improving our paper!

  1. Methods: Section 3.1: Some parts of the mass spectroscopy methods were not included (solvents, flow rates, ionization energy, etc.). Which solvent systems were used when determining the Rf values?

Response 2. As most of the requested details are given in the headers of the mass spectra, we inserted just only a citation into the chapter "General Procedures". The TLC Rf-values were inserted as a footnote in Table 1

  1. The conclusion (lines 301-302) states that the crude extract was strongly active against viridochromogenes as described. Do you have ideas for why the activity was greater in the crude extract than in purified metabolites? It might be expected that a more purified component would show more activity.

Response 3. This is absolutely true, if the isolated metabolites are responsible for the activity of the crude extract. In this case, a decreasing activity during isolation may indicate synergistic effects, which is, however, not often true. We are assuming therefore, that a) highly active trace compounds hat got lost during purification (selective adsorption/absorption, or that b) unstable compounds were lost due to hydrolysis, oxidation, re-arrangement or any other structural change associated with a loss of activity. This is a frequently observed problem, where work-up would require a different approach for the isolation of the active metabolites. In case b), frequently stable artifacts are found (e.g. benzene derivatives from en-diynes etc.), which are giving hints for the original bio-active structures. However, as the activity of our crude extract was not very stimulating, we did not further follow this idea.

Reviewer 2 Report

The manuscript entitled "New Haloterpenes from the Marine Red Alga Laurencia papillosa: Structure Elucidation and Biological Activity" describes the isolation and structure elucidation of 6 nominally new halogenated terpenoids from a marine alga. Absolute stereochemical configuration was determined unambiguously for two of the compounds using x-ray crystallography, and the remaining absolute configurations were inferred based on the known ones (and presumed same biosynthetic origins), and on the basis of DFT calculations. A simple agar diffusion assay was also performed on a collection of bacterial strains, revealing modest activity for two of the known molecules (10 mm inhibition at 40 ug/disk). The content, techniques, and presentation is all pretty standard, with nothing especially novel or innovative. However, the science described is sound, and the new structures and biological activity deserve to have a place in the literature. As a result, I recommend this manuscript for publication in Marine Drugs.

Comments:

  1. Is Marine Drugs published in American English? There are a few awkward phrases that I’m guessing are a translation thing, like the references to “digits after comma” in the Table 3 legend.
  2. I had trouble following the naming conventions. The name of compound 5a struck me as particularly confusing. Is dihydro the right modifier, when the groups added in place of the additional double bonds are both hydroxyl groups, not hydrogen atoms? (Perhaps so, but I’d just want to double check it.)
  3. Also, given the names that the authors propose for the new molecules, I assumed there must be such a thing as aplysiol in the known literature, but for some reason the authors don’t actually mention the existence of this family of compounds at all in the introduction. It isn’t until page 4, line 109 that their existence (or at least that of aplysiadiol) is acknowledged at all, which seemed odd to me. The relevant family of compounds seems like context that would be appropriate for the introduction (if not the abstract).
  4. Similarly, although structure 4 appears in Figure 1, it isn’t until page 4 that it is referenced at all, in the context of absolute configuration.
  5. Table 1 was interesting. It is unusual to see TLC staining behavior figuring prominently in the body of a manuscript these days, although honestly I kind of liked it. I did also wonder why IR data was only included for compound 6.
  6. I found figures 2-8 to be pretty off-putting for two major reasons:
    • There are way too many arrows! I appreciate the desire to be comprehensive, but would suggest moving the exploding arrows to the SI, and using the in-body figure to highlight just the most important correlations for determining the structure. Another option is to catalog all the correlations in a neat NMR table. (Tables 2 and 3 could be modified to include HMBC and COSY columns, for example.)
    • In addition, the figures are inconsistent, with different fonts, font sizes (for both atom labels and chemical shifts). Carbon chemical shifts are labeled in some figures and not others, and some figures have wedges but not others. It’s distracting.

Author Response

Thank you for your kind recommendation and for the very helpful comments and suggestions!

Point 1: Is Marine Drugs published in American English? There are a few awkward phrases that I’m guessing are a translation thing, like the references to “digits after comma” in the Table 3 legend.

Response 1. We discussed this special topic with a native American and corrected it into "digits behind the decimal point". On request, we are open to send the whole manuscript a second time for language polishing, but this would need again 3-4 weeks. Anyway, if it is needed and cannot be done in the editorial office, we will do it of course.

Point 2: I had trouble following the naming conventions. The name of compound 5a struck me as particularly confusing. Is dihydro the right modifier, when the groups added in place of the additional double bonds are both hydroxyl groups, not hydrogen atoms? (Perhaps so, but I’d just want to double check it.)

Response 2: We are fully agreeing! To apply the IUPAC rules on a trivial name is normally restricted to common chemicals like acetic acid, cholesterol etc., were the systematic name would be too strange or too long. We discarded the option to generate a new name of its own and decided to use a group of related compounds as basis. A specialist on red algae may (hopefully) see now immediately, to which group of natural products our compounds are belonging.

On basis of aplysiadiol (4), a 11,14-dihydro derivative would have a D12 double bond as in 5a, and a additional hydroxylation at C-11,14 would end up in our suggested name, which follows the same logic as aplysiadiol for 4. In the same way, the name "anhydroaplysiadiol" (Takahashi, Y; Suzuki, M; Abe, T; Masuda, M. Phytochemistry (1998), 48(6), 987-990) was used for a derivative of 4 with an additional D15 double bond, formed by loss of water at C-15.

The problem is, that no trivial name has been published for the (known) hydrocarbon "1-debromo-5,15-dehydroxy-4", and this seems to be the confusing observation you mentioned. But what else can we do? All suggestions would be very welcome.

Point 3: Also, given the names that the authors propose for the new molecules, I assumed there must be such a thing as aplysiol in the known literature, but for some reason the authors don’t actually mention the existence of this family of compounds at all in the introduction. It isn’t until page 4, line 109 that their existence (or at least that of aplysiadiol) is acknowledged at all, which seemed odd to me. The relevant family of compounds seems like context that would be appropriate for the introduction (if not the abstract).

Response 3: In the revised version of the manuscript, we have referred to that in the introductory part. We are thankful for this remark!

Point 4: Similarly, although structure 4 appears in Figure 1, it isn’t until page 4 that it is referenced at all, in the context of absolute configuration.

Response 4: Please see our response in point 3

Point 5: Table 1 was interesting. It is unusual to see TLC staining behavior figuring prominently in the body of a manuscript these days, although honestly, I kind of liked it. I did also wonder why IR data was only included for compound 6.

Response 5: We omitted the IR spectra of the other compounds for two reasons: 1) Only for compound 6, the IR spectrum was helpful for the structure elucidation. 2) Compound 3 was containing 10-hydroxykahukuene B as impurity, and 5 was a mixture of epimers.

Point 6: I found figures 2-8 to be pretty off-putting for two major reasons:

Point 6a: There are way too many arrows! I appreciate the desire to be comprehensive but would suggest moving the exploding arrows to the SI and using the in-body figure to highlight just the most important correlations for determining the structure. Another option is to catalog all the correlations in a neat NMR table. (Tables 2 and 3 could be modified to include HMBC and COSY columns, for example.)

 Response 6a: As suggested, we moved the referred figures to the SI and just inserted the important correlations of H,H COSY and HMBC correlations for compounds 1-3and 5a in Figure 2, and those for compound 6 (along with its NOESY correlations) in Figure 5. So, totally, we have reduced the figures to 6 instead of 9 in the orignal manuscript. On the other hand, we have inserted the detailed H,H COSY and HMBC correlations for the referred compounds in SI as Figures S52-S55, and we mentioned that at the end of the working up section (3.3. Extraction and isolation of the bioactive constituents).

Point 6b: In addition, the figures are inconsistent, with different fonts, font sizes (for both atom labels and chemical shifts). Carbon chemical shifts are labeled in some figures and not others, and some figures have wedges but not others. It’s distracting.

Response 6b: Thank you, the fonts and sizes of structures in the referred figures have been revised and should be consistent now.

Reviewer 3 Report

This work by Shaaban et al. characterizes the isolation and structure elucidation of six metabolites from the red algae Laurencia papillosa. The paper is exceedingly well-written, the 1D and 2D NMR experiments are rigorously described and depicted in figures and tables, and the X-ray crystallography studies add additional confidence in the structure assignments. This is another strong piece of work from the Professor Dr. Laatsch group concerning identification of structurally interesting natural products. This paper should be accepted for publication in Marine Drugs with very minor changes.

I have only minor comments.

Page 4 of 13, Line 97 = Change "easily" to "easy"

Page 5 of 13, Line 123 = Change "a noticeable less polarity" to "with noticeably less polarity"

Author Response

Point 1: Page 4 of 13, Line 97 = Change "easily" to "easy"

Response 1: OK, was done

Point 2: Page 5 of 13, Line 123 = Change "a noticeable less polarity" to "with noticeably less polarity"

Response 2: was done, thank you so much
